# Machine Learning Benchmark on Dynamic Functional Connectivity: Promise, Pitfalls, and Interpretations

## Abstract

An unprecedented amount of existing functional Magnetic Resonance Imaging (fMRI) data provides a new opportunity to understand the relationship between functional fluctuation and human cognition/behavior using a data-driven approach. To that end, tremendous efforts have been made in machine learning to predict cognitive states from evolving volumetric images of blood-oxygen-level-dependent (BOLD) signals. Due to the complex nature of brain function, however, the evaluation on learning performance and discoveries are not often consistent across current state-of-the-arts (SOTA). By capitalizing on large-scale existing neuroimaging data (39,784 data samples from seven databases), we seek to establish a well-founded empirical guideline for designing deep models for functional neuroimages by linking the methodology underpinning with knowledge from the neuroscience domain. Specifically, we put the spotlight on (1) What is the current SOTA performance in cognitive task recognition and disease diagnosis using fMRI? (2) What are the limitations of current deep models? and (3) What is the general guideline for selecting the suitable machine learning backbone for new neuroimaging applications? We have conducted a comprehensive evaluation and statistical analysis, in various settings, to answer the above outstanding questions. In addition, we explore a novel attention learning mechanism to provide meaningful spatial pattern of brain activation that is associated with various cognitive tasks and neurological conditions.

## 1 Introduction

Functional magnetic resonance imaging (fMRI) is a popular non-invasive neuroimaging technology extensively used to investigate brain activity (Smith et al., 2004; Van Dijk et al., 2004; Fox et al., 2005) and diagnosis neurological disorders (Klöppel et al., 2008; Zhou et al., 2008; Khazaee et al., 2015; Deshpande et al., 2013; Calhoun et al., 2004). Monitoring alterations in blood oxygen level-dependent (BOLD) signal, an indicator of local blood flow and oxygenation changes in response to neural activity, offers a window into the functional activity of the brain *in-vivo* (Biswal et al., 2010). During fMRI scans, subjects typically engage in either task-related activities to measure neural activity associated with specific cognitive tasks, or resting states to measure spontaneous fluctuations supported by intrinsic neural activity.

**Glance of deep learning on fMRI.** As shown in Fig. 1 left, an fMRI scan consists of a set of evolving volumetric brain mapping of BOLD signal over time (i.e., $3D + t$). Instead of using 4D image directly, it is a common practice to partition the whole brain into a set of parcellations (by following a pre-defined atlas (Evans et al., 2001)) and analyze the mean-time course of BOLD signals. In this regard, sequential models such as Recurrent Neural Networks (RNNs)(Yan et al., 2019) and Transformer architectures(Bedel et al., 2023; Thomas et al., 2022) have been adapted to map time series to the phenotypic traits such as cognitive status or diagnostic label.

Meanwhile, synchronized functional fluctuations between distinct brain regions, a.k.a. functional connectivity (FC), has been discovered in many neuroscience studies (Biswal et al., 1995; Friston et al., 1994; Sporns et al., 2005; Greicius et al., 2003). This discovery shifted research focus to investigating region-to-region interactions, giving rise to a new interdisciplinary field — network

neuroscience, which conceptualizes the brain as a connectome: an interactive network map where distinct brain regions synchronize their neural activities through myriad interconnecting nerve fibers and functional co-activations (Bassett & Sporns, 2017; Park & Friston, 2013; Bassett & Bullmore, 2006). Since connectome is often encoded in a graph structure, graph neural networks (GNNs) come to the stage for capturing putative graph representations for nodes and edges in the FC brain network (Han et al., 2020; Wang et al., 2021; Zhu et al., 2021). In contrast to conventional GNNs, current deep models for FC predominantly focus on graph classification tasks, that is, assigning a label to a graph using integrated feature representations of the brain network.

For simplicity, many deep models for FC assume that brain connectivities do not change while scanning a resting-state fMRI. However, there is a growing consensus in the neuroimaging field that spontaneous fluctuations and correlations of signals between two distinct brain regions change with correspondence to cognitive states, even in a task-free environment (Allen et al., 2014; Bassett et al., 2011; Calhoun et al., 2014; Hutchison et al., 2013). Thus, it is natural to perform graph inference in the temporal domain by combining time series modeling and graph representation learning techniques. For example, as FC is a symmetric and positive-definite (SPD) matrix on the Riemannian manifold (Dan et al., 2022b), a manifold-based deep model is trained to capture graph spatial features while the evolution of FC matrices is jointly modeled by an RNN.

**Challenges for enhancing the applicability of current deep models.** Despite tremendous efforts that have been made to develop state-of-the-art deep models for fMRI, there is a significant gap between the general model design and diverse application scenarios in both neuroscience and routine clinical practice, due to the following challenges. *First*, most deep models are evaluated in the limited (in-house) datasets only. Such gross simplification is responsible for lacking comprehensive and trustful benchmarks across various deep models. *Second*, little attention has been paid to enhancing model interpretability despite the high demand for an in-depth understanding of how anatomical structures support brain function and how self-organized functional fluctuations give rise to cognition and behavior. *Third*, current works mainly focus on "one-size-fits-all" solution for all fMRI studies. Considering the complex nature of brain functions in different environments, we hypothesize that the best practice for real-world problems is to deploy the most suitable models tailored to specific characteristic of data and applications. To that end, it is critical to examine this hypothesis and further gain conclusive insights through existing large-scale public fMRI benchmarks.

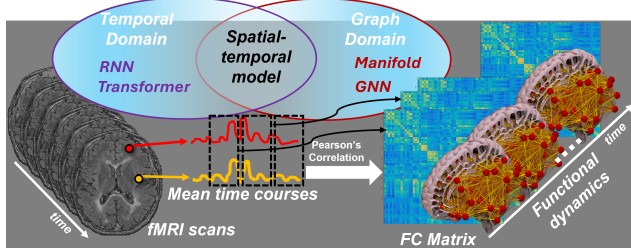

Figure 1: Various data representations in fMRI studies (in time series or graph) and machine learning models for fMRI analyses.

**Our contributions.** The researchers in this field proposed a number of deep learning approaches for graph feature representation learning (Sim et al., 2024; Park et al., 2023; Baek et al., 2023; Choi et al., 2022; Ma et al., 2021; Gan et al., 2020; Dan et al., 2023), dynamic network analysis (Cho et al., 2023; Turja et al., 2023; Dan et al., 2022d; Huang et al., 2021; Yang et al., 2021; Zhen et al., 2021; Ma et al., 2020; Dan et al., 2022b), and computer-assisted disease early diagnosis (Cho et al., 2024; Shi et al., 2024; Dan et al., 2022a; Xiao et al., 2022; Yang et al., 2022) using human connectome data. By capitalizing on the extensive experience of developing machine learning techniques for neuroimaging studies and unprecedented amount of existing high-quality public functional neuroimages (summarized in Table 1), we seek to provide a comprehensive report on *model performance* and *model explainability* of various contemporary solutions across different fMRI scenarios (covering both task-evoked fMRI and disease-related resting-state fMRI). The specific contributions of this paper are as follows:

1. We have constructed, pre-processed and released a diverse set of datasets for the fMRI-based benchmark (see Table 1), including the Human Connectome Project (HCP-YA and HCP-Aging) (Van Essen et al., 2013) dataset for task classification, the ADNI (Mueller et al., 2005) and OASIS clamontagne2019oasis datasets for Alzheimer's disease diagnosis, the PPMI dataset for Parkinson's disease diagnosis, and the ABIDE dataset for Autism diagnosis (Xu et al., 2024) (PPMI and ABIDE are from (Xu et al., 2024)).

2. We have trained and evaluated an inclusive set of popular spatial models (using topological heuristics from functional connectivity) and sequential models (using temporal dynamics from BOLD signal) for fMRI classification across all datasets. We thoroughly discussed the model performances in different fMRI learning scenarios and investigated the underlying reasons for their performance differences in the neuroscience perspective by in-depth remarks, which would be beneficial for engineering suitable deep models for new applications.

3. We have particularly investigated a post-hoc method to generate task-specific attention maps of models and evaluated neural activation patterns across different tasks. This exploration aims to uncover the neurobiological mechanism using data-driven approaches, which is pivotal for promoting machine learning techniques in computational neuroscience and clinical applications.

## 2 RELATED BENCHMARK ON COMPUTATIONAL FMRI ANALYSIS

Several benchmark studies have been conducted on fMRI data. For instance, ElGazzar et al. (2022) evaluated the performance of various GNNs on the UK Biobank database, discussing different node features and adjacency matrices for different GNNs, and conducted experiments on both static and dynamic FCs. Similarly, Said et al. (2023) provided a more comprehensive discussion including node feature types, the number of ROIs, and adjacency matrix sparsity, as well as identified optimal settings for these parameters. They evaluated 12 models but on the HCP dataset only. Xu et al. (2024) curated and released six disease-based datasets and assessed 13 various machine learning methods on these datasets.

Despite these efforts, to the best of our knowledge, no benchmark work focuses on the general principle of designing/tailoring suitable models for various fMRI studies. To address this, our work includes a detailed analysis of the HCP dataset, which features both single-domain and multi-domain cognitive tasks, as well as the ADNI, OASIS, and PPMI datasets for neurodegenerative diseases, and the ABIDE dataset for neuropsychiatric disorders. Additionally, recent works (Bedel et al., 2023; Thomas et al., 2022) have demonstrated the efficiency of sequential models, such as Transformers, in fMRI analysis. Therefore, our study not only examines spatial models based on FC but also extensively evaluates sequential models leveraging temporal information, offering a more comprehensive benchmark in the field.

## 3 DATASETS

As shown in Table 1, we uses a diverse array of datasets (a total of 39,784 fMRI images), including the Human Connectome Project (HCP-YA and HCP-Aging) dataset (with various cognitive tasks), the Alzheimer's Disease Neuroimaging Initiative (ADNI), the Open Access Series of Imaging

Table 1: The Summarization of Benchmarking Datasets.

| Dataset | # of samples | # of class | mean of length | # of ROIs | # of edges |
|---|---|---|---|---|---|
| HCP-Task | 14,860 | 7 | 278 | 360 | 12,960 |
| HCP-WM | 17,296 | 8 | 39 | 360 | 12,960 |
| HCP-Aging | 4,897 | 4 | 334 | 116 | 1,346 |
| ADNI | 250 | 2 | 177 | 116 | 1,346 |
| OASIS | 1,247 | 2 | 390 | 160 | 2,560 |
| PPMI | 209 | 4 | 198 | 116 | 1,346 |
| ABIDE | 1,025 | 2 | 200 | 116 | 1,346 |

Studies (OASIS), the Parkinson's Progression Markers Initiative (PPMI), and the Autism Brain Imaging Data Exchange (ABIDE). The description about MRI scanner and imaging protocol is summarized in Appendix A.1.1.

## 4 BENCHMARKING MODELS

The selection of models for benchmarking in fMRI analysis, as shown in Fig. 2, is driven by the need to represent the multivariate nature of fMRI. They fall into two main categories: spatial models and sequential models, offering advantages in analyzing brain connectivity and temporal dynamics, respectively.

**Spatial models** are crucial for understanding the brain functional connectivity patterns. Traditional GNNs like Graph Convolutional Network (GCN) (Kipf & Welling, 2016) and Graph Isomorphism Network (GIN) (Xu et al., 2018) are selected because they effectively represent the diffusion pattern

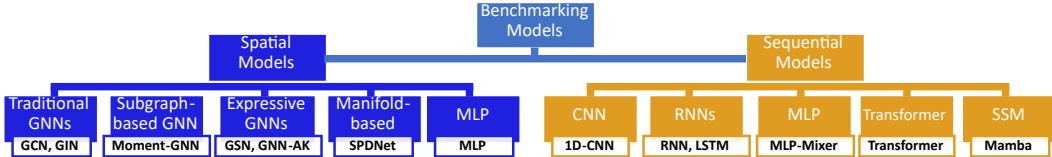

Figure 2: Selected deep models for the benchmark, which include spatial (left) and sequential models (right).

and isomorphism encoding for brain connectivity. Subgraph-based GNNs, such as Moment-GNN (Kanatsoulis & Ribeiro, 2023), focus on subgraph structures within the brain network. This allows for the identification of localized patterns that might be missed by traditional GNNs. Expressive GNNs, including Graph Substructure Network (GSN) (Bouritsas et al., 2022) and GNNAsKernel (GNN-AK) (Zhao et al., 2021), are chosen for their enhanced expressivity by incorporating the counting of subgraph isomorphisms and local subgraph encoding respectively, which may be beneficial for distinguishing subtle differences in FC. A manifold-based model like Symmetric Positive Definite Network (SPDNet) (Dan et al., 2022c) is adopted for its capability to handle high-dimensional manifold data, making it effective for high-resolution fMRI. A traditional Multi-Layer Perceptron (MLP) serves as a baseline model for its high efficiency and versatility.

**Sequential models** are selected for analyzing the temporal dynamics of BOLD signals. 1D-CNN is selected for its strength in capturing temporal patterns through convolutional operations. Recurrent Neural Network (RNN) (Rumelhart et al., 1986) and Long Short-Term Memory (LSTM) (Hochreiter & Schmidhuber, 1997) are included for their proficiency in modeling sequential BOLD signal and capturing long-range dependencies. MLP-Mixer (Tolstikhin et al., 2021) is chosen for mixing spatial and temporal features, offering a comprehensive view of BOLD by integrating information across different dimensions. Transformer (Vaswani et al., 2017) is selected for its powerful attention mechanisms, which enable it to capture global dependencies in the BOLD signals. Finally, State-Space Model (SSM) represented by Mamba (Gu & Dao, 2023) is selected for its advanced state-space modeling capabilities that capture the dynamics of brain activity over time.

## 5 EVALUATE THE MODEL PERFORMANCE ON FMRI DATA

We seek to examine the following hypotheses through a total of 13 benchmark deep models on 39,784 fMRI images from seven data cohorts. (The workflow of fMRI image pre-processing is summarized in Appendix A.1.2.)

*(H1) Are there any deep models that consistently perform well across all fMRI studies?*

*(H2) Is there a connection between the design of deep models and the underlying biological question?*

*(H3) What is the general guideline for developing fMRI-based deep models for new neuroscience and clinical applications?*

*(H4) What is the clinical value of fMRI-based deep models in disease diagnosis?*

In the following experiments, we focus on an analysis of the relationship between the learning mechanisms of model backbones and their application scenarios in a neuroscience context. To that end, we assume that other confounding variables, such as the choice of atlas [1] and multi-site issues [2], have been professionally addressed.

### 5.1 EXPERIMENTAL SETUP

Suppose the whole brain BOLD signals are represented as $\mathbf{X} \in \mathbb{R}^{N \times T}$, where $N$ and $T$ denote the number of brain parcellations and time points. Let $\mathbf{W} = [w_{ij}]_{i,j=1}^{N} \in \mathbb{R}^{N \times N}$ be a FC matrix

---

[1]The effect of atlas parcellations on model performance has been evaluated in (Said et al., 2023). The general conclusion is that fine-grained brain parcellation often leads to higher prediction accuracy of phenotypical traits using fMRI data. Thus, we use the Human Connectome Project (HCP) multi-modal parcellation of the human cerebral cortex (Glasser et al., 2016) which is a popular fine-grained atlas with 360 distinct regions based on a combination of anatomical, functional, and connectivity data.

[2]Multi-site issue refers to the external variations (related to scanner and imaging protocol) that arise when data are collected from multiple research sites or centers. Thus, data harmonization has been applied to the fMRI data in our work.

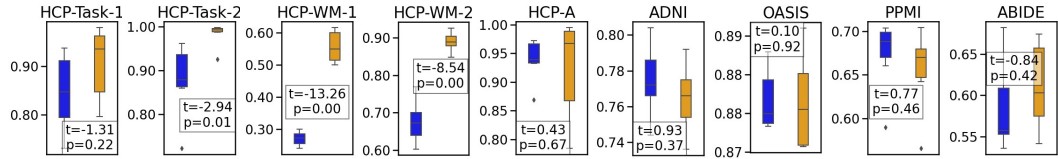

Figure 3: Significant tests between spatial models (■) and sequential models (■) for various datasets. The first and third plots for the HCP dataset are "Separated Scan 1 & Scan 2" experiment (marked as '###-1'), and the second and fourth plots are "Mixed Scan 1 & Scan 2 setting" (marked as '###-2').

constructed using Pearson's correlation as $\mathbf{W} = \mathbf{X}\mathbf{X}^\intercal$. As suggested in (Bedel et al., 2023), we retain the top 10% of values in FC matrix $\mathbf{W}$ based on the degree of FC and truncate the rest to 0 to enhance the sparsity in the brain network.

For spatial models, the input includes (1) FC matrix $\mathbf{W}$ that determines the graph diffusion process and (2) the node embedding vectors that describe the local topology at each node. Following the optimal settings described in (Bedel et al., 2023), we use the vector of FC connectivity degree $\mathbf{w}_i = [w_{ij}]_{j=1, j\neq i}^N$ as the initial feature embedding for the node $v_i$.

For sequential models, it is important to standardize the input BOLD signal $\mathbf{X}$ with varying time length to uniform dimensions. In datasets where BOLD signals are recorded with varying time points, we employ zero-padding to achieve this alignment (see Appendix A.2 for details).

Each participant in HCP-Task has two scans, Scan 1 (i.e., test fMRI) and Scan 2 (i.e., re-test fMRI) [3], which have same task events in different orders. To ensure the replicability of the experiment, we implemented two distinct experimental settings accordingly. For "Separated Scan 1 & Scan 2" scenario, models were trained on Scan 1 only and validated and tested on Scan 2. This setting not only evaluates the classification performance but also assesses the model's replicability across different scans representing the same task. For "Mixed Scan 1 & Scan 2" scenario, we mixed both scans, using half for training and the remaining half for validation and testing.

## 5.2 ANALYSIS ON TASK-EVOKED FMRI

*First*, we benchmark on task recognition for seven cognitive tasks (denoted by HCP-Task): emotion, relational, gambling, language, social, motor, and working memory. The scan duration of each task is ranging from 176 to 405 time points ($0.72 \times 176 \approx 127$ seconds - $0.72 \times 405 \approx 292$ seconds). *Second*, we test the task recognition accuracy for HCP-Working Memory (HCP-WM) data where each fMRI scan consists of eight short-term tasks: 0bk-(place, tools, face, body) and 2bk-(place, tools, face, body). The scan duration in HCP-WM is relatively short with only 39 time points ($0.72 \times 39 \approx 28$ s). Last, we benchmark on the task recognition for three PsychoPy tasks covered in the HCP-A dataset include VisMotor Task, CARIT (Conditioned Approach Response Inhibition) Task and Facename Task.

**Sequential vs. spatial model comparison in HCP-Task.** As shown in Table 2, both spatial and sequential models demonstrate a wide variety in task recognition accuracy, with sequential models generally achieving higher accuracy levels. Additionally, the statistical analysis of boxplots in Figure 3 reveals that sequential models tend to have higher accuracy scores, as evidenced by their distribution towards the upper end of the scale. Notably, there is a significant difference ($p = 0.01$) between the two types of models in the Mixed Scan setting.

**Sequential vs. spatial model comparison in HCP-WM.** In Table 2, sequential models not only outperform spatial models but do so by a considerable margin, showcasing a performance advantage of up to 30% in accuracy across both settings. This performance gain is further supported by a $t$-statistic of $-13.3$ and $-8.54$ at a significance level of $p < 10^{-4}$ in two settings, indicating that the performance difference in this task is both substantial and statistically significant.

**Sequential vs. spatial model comparison in HCP-Aging.** In Table 2 bottom, most of the sequential models can achieve higher accuracy than the spatial models, especially 1D-CNN and Mamba, which have exceeded 99% in Acc, precision and F1. This indicates that the sequential models' effectiveness on this dataset. However, as illustrated in Fig.3, while sequential models demonstrate better perfor-

---

[3]In many fMRI studies, test-retest fMRI scans involve acquiring multiple sessions of fMRI from the same individuals, allowing researchers to assess the reliability and reproducibility of brain activation patterns.

mance on this dataset, the variability in accuracy among these models is higher. Consequently, the advantage of sequential models over spatial models is not statistically significant, as evidenced by the broader range of Acc differences.

**Remarks 1.1.** One possible explanation, from *the perspective of machine learning*, for the enhanced performance of sequential models compared to spatial models could be attributed to a stronger correlation between the dynamic characteristics of BOLD signals and cognitive tasks, as opposed to the correlation with (static) wiring topology of functional connectivities in healthy brains. The evidence for supporting this statement becomes particularly more apparent in HCP-WM experiment. The *biological basis* for the differences in performance between sequential and spatial models lies in the nature of task-evoked experiments. These experiments are designed to elicit specific brain responses related to cognitive processes such as attention, memory, language, or emotion through simple cognitive tasks or stimuli. In this context, some portion of the brain exhibits higher activity levels, as evidenced by the fluctuation dynamics in BOLD signals, to support the targeted cognitive tasks. Meanwhile, the human brain exhibits limited possibility to reconfigure the topology of whole-brain functional connectivities as mounting evidence shows that network organization remains stable regardless of brain states switching from task to task (Buckner et al., 2009; 2013; Power et al., 2011).

**Remarks 1.2.** Sequential models are more effective in task fMRI classification, and there exist significant variations of learning performance within spatial and sequential models. The evidence supporting this remark is discussed method-to-method in Appendix A.4.1.

Table 2: Performance on task-evoked fMRI. Blue is best performance in spatial models, and Orange is best performance in sequential models.

| | Spatial Models | | | | | | | Sequential Models | | | | | |
|---|---|---|---|---|---|---|---|---|---|---|---|---|---|
| (%) | GCN | GIN | GSN | MGNN | GNN-AK | SPDNet | MLP | 1D-CNN | RNN | LSTM | Mixer | TF | Mamba |
| HCP-Task (Separated Scan 1 & Scan 2) | | | | | | | | | | | | | |
| Acc | 84.72 | 84.50 | 88.78 | 74.18 | 73.01 | 93.76 | 93.54 | 97.96 | 79.54 | 82.35 | 96.48 | 91.57 | 95.60 |
| Pre | 85.30 | 84.60 | 88.92 | 76.30 | 73.00 | 93.74 | 93.96 | 97.98 | 78.32 | 84.07 | 96.54 | 92.11 | 95.69 |
| F1 | 84.82 | 84.46 | 88.79 | 73.98 | 72.99 | 93.70 | 93.52 | 97.95 | 78.79 | 80.07 | 96.37 | 90.80 | 95.58 |
| HCP-Task (Mixed Scan 1 & Scan 2) | | | | | | | | | | | | | |
| Acc | 87.86 | 86.59 | 92.54 | 86.00 | 72.19 | 96.23 | 94.62 | 98.74 | 92.51 | 99.51 | 99.78 | 99.60 | 99.06 |
| Pre | 87.95 | 86.89 | 92.75 | 86.14 | 73.13 | 96.26 | 94.63 | 98.75 | 92.56 | 99.51 | 99.78 | 99.60 | 99.06 |
| F1 | 87.87 | 86.59 | 92.58 | 85.96 | 72.87 | 96.28 | 94.60 | 98.74 | 92.51 | 99.51 | 99.78 | 99.60 | 99.06 |
| HCP-Working Memory (Separated Scan 1 & Scan 2) | | | | | | | | | | | | | |
| Acc | 26.83 | 27.23 | 30.06 | 24.05 | 24.07 | 29.54 | 27.60 | 49.94 | 61.55 | 61.46 | 54.38 | 50.38 | 55.58 |
| Pre | 23.54 | 23.92 | 29.68 | 18.40 | 18.56 | 29.86 | 25.77 | 48.13 | 63.00 | 62.94 | 52.59 | 49.62 | 57.09 |
| F1 | 23.55 | 23.46 | 27.43 | 19.41 | 19.66 | 26.42 | 25.71 | 48.14 | 60.92 | 60.83 | 52.52 | 48.69 | 54.45 |
| HCP-Working Memory (Mixed Scan 1 & Scan 2) | | | | | | | | | | | | | |
| Acc | 68.97 | 62.07 | 71.18 | 65.72 | 60.31 | 67.26 | 76.91 | 89.90 | 90.67 | 84.87 | 87.88 | 87.90 | 92.59 |
| Pre | 69.13 | 63.16 | 71.35 | 67.46 | 61.25 | 67.35 | 76.87 | 89.90 | 90.70 | 85.00 | 87.93 | 87.94 | 92.61 |
| F1 | 68.98 | 62.10 | 71.16 | 65.93 | 60.33 | 67.34 | 76.86 | 89.89 | 90.68 | 84.89 | 87.88 | 87.88 | 92.60 |
| HCP-Aging | | | | | | | | | | | | | |
| Acc | 93.85 | 93.40 | 96.45 | 93.26 | 86.93 | 96.77 | 97.2 | 99.47 | 78.44 | 83.79 | 97.92 | 95.61 | 99.14 |
| Pre | 93.95 | 93.42 | 96.53 | 93.29 | 86.44 | 96.80 | 97.25 | 99.47 | 78.79 | 84.82 | 97.93 | 95.90 | 99.15 |
| F1 | 93.85 | 93.37 | 96.46 | 93.23 | 85.96 | 96.77 | 97.21 | 99.47 | 77.54 | 83.49 | 97.91 | 95.66 | 99.14 |

## 5.3 Analysis on Early Diagnosis of Neurodegenerative Diseases Using Resting-State fMRI

**Early diagnosis for neurodegenerative diseases.** We take AD and PD as examples due to the abundance of disease-related public data cohorts. In the following, we examine the classification performance between cognitive normal (CN) and subjects with neurodegenerative disease (ND) using resting-state fMRI data.

**AD classification on ADNI.** In Table 3, we show the accuracy, precision, and F1-score for CN vs. AD classification by spatial models (left) and sequential models (right) [4]. It is shown that the spatial models perform slightly better than the sequential models across the most of evaluation metrics. Furthermore, as shown in Figure 3, at a significance level $p = 0.37$, spatial models exhibit no significant performance improvement over sequential models in a two-sample $t$-test ($t$-statistic: 0.93). Therefore, the overall performance of spatial models exceeds that of sequential models, there are still some variations within spatial models.

**AD classification on OASIS.** The CN vs. AD classification results in Table 3 indicate a comparable performance between sequential models and spatial models, as the accuracy distribution for both types of models is relatively tight around the 0.873 to 0.886 range. Both types of model show a consistent performance across different metrics. In addition, there is no statistically significant difference in accuracy between the two model types.

**PD classification on PPMI** Table 3 presents a comprehensive performance evaluation on identifying CN, SWEDD (scans without evidence of dopaminergic deficit), Prodromal, and PD subjects (four-class classification) across multiple models. In general, spatial models' performance is slightly better than sequential models as only four of seven spatial models are generally higher than those of their sequential counterparts except for Transformer. Also, there is no statistical difference in accuracy between sequential and spatial models (visually confirmed by the box plot in Fig. 3).

Table 3: Performance on neurodegenerative disease fMRI. Blue is best performance in spatial models, and Orange is best performance in sequential models.

| (%) | Spatial Models | | | | | | | Sequential Models | | | | | |
|---|---|---|---|---|---|---|---|---|---|---|---|---|---|
| | GCN | GIN | GSN | MGNN | GNN-AK | SPDNet | MLP | 1D-CNN | RNN | LSTM | Mixer | TF | Mamba |
| **ADNI** | | | | | | | | | | | | | |
| Acc | 74.40 ±3.67 | 76.40 ±6.05 | 79.20 ±4.66 | 76.80 ±3.92 | 77.20 ±6.21 | 78.50 ±5.73 | 80.40 ±4.54 | 76.00 ±6.45 | 75.20 ±6.14 | 77.60 ±6.25 | 77.20 ±5.95 | 79.20 ±5.31 | 73.60 ±5.12 |
| Pre | 67.12 ±12.30 | 69.75 ±16.55 | 82.37 ±4.81 | 76.80 ±9.67 | 75.52 ±13.41 | 65.04 ±9.01 | 81.38 ±5.55 | 72.92 ±14.98 | 69.66 ±13.88 | 76.11 ±13.32 | 77.87 ±12.76 | 78.53 ±10.50 | 71.53 ±12.23 |
| F1 | 67.52 ±5.87 | 69.61 ±9.92 | 75.75 ±4.92 | 72.49 ±6.08 | 71.46 ±9.81 | 61.91 ±13.62 | 78.46 ±4.99 | 68.99 ±9.60 | 68.90 ±8.94 | 72.48 ±9.05 | 72.09 ±9.56 | 75.39 ±7.58 | 67.85 ±8.80 |
| **OASIS** | | | | | | | | | | | | | |
| Acc | 88.13 ±1.04 | 87.49 ±0.93 | 87.33 ±1.27 | 87.41 ±0.79 | 87.73 ±1.64 | 88.29 ±0.8 | 87.33 ±1.27 | 88.59 ±0.97 | 87.07 ±1.06 | 87.07 ±1.06 | 87.15 ±0.95 | 88.03 ±1.49 | 87.95 ±2.77 |
| Pre | 87.44 ±2.56 | 84.90 ±4.75 | 78.08 ±5.21 | 84.28 ±5.37 | 89.27 ±1.21 | 55.84 ±4.51 | 77.93 ±4.92 | 89.16 ±1.13 | 75.82 ±1.85 | 75.82 ±1.85 | 77.69 ±2.74 | 85.58 ±5.17 | 87.51 ±2.91 |
| F1 | 84.26 ±1.40 | 82.40 ±1.03 | 81.73 ±2.22 | 81.84 ±0.89 | 82.58 ±2.31 | 56.47 ±7.42 | 81.83 ±2.37 | 84.59 ±1.63 | 81.05 ±1.51 | 81.05 ±1.51 | 81.71 ±1.08 | 83.61 ±2.90 | 83.92 ±4.02 |
| **PPMI** | | | | | | | | | | | | | |
| Acc | 68.02 ±11.57 | 70.33 ±8.72 | 70.40 ±12.48 | 69.45 ±10.37 | 68.83 ±7.70 | 66.02 ±10.10 | 58.98 ±10.94 | 68.02 ±10.75 | 56.55 ±7.21 | 64.21 ±10.56 | 66.12 ±11.03 | 70.43 ±11.74 | 67.93 ±10.69 |
| Pre | 60.28 ±18.09 | 66.64 ±11.05 | 70.63 ±14.00 | 63.10 ±15.32 | 63.26 ±11.75 | 42.92 ±15.25 | 62.43 ±13.15 | 65.33 ±13.50 | 45.15 ±15.34 | 57.86 ±18.23 | 63.27 ±16.97 | 66.59 ±13.26 | 66.40 ±11.44 |
| F1 | 61.56 ±15.25 | 64.84 ±10.62 | 66.95 ±13.64 | 63.23 ±13.29 | 63.76 ±9.01 | 40.14 ±17.60 | 57.84 ±11.82 | 61.41 ±13.42 | 43.14 ±8.46 | 56.25 ±15.00 | 58.64 ±14.44 | 64.68 ±14.35 | 59.11 ±8.87 |

**Remarks 2.1.** In contrast to task-evoked fMRI, which captures brain activity triggered by specific tasks, resting-state fMRI reflects spontaneous brain activity in the absence of tasks, with BOLD signals indicating intrinsic functional connectivity between brain regions. Due to such significant differences in biological mechanisms, spatial models exhibit higher classification accuracy than sequential models. Meanwhile, some neurological impairment may reflect dysfunction rather than loss of neurons in neurodegenerative diseases (Palop et al., 2006; Pievani et al., 2014; Matthews et al., 2013). In this regard, ND can be understood as a disconnection syndrome where the large-scale brain network is progressively disrupted by neuropathological processes (Chiesa et al., 2017). The evidence supporting this remark shown in Table 3 and Fig. 3 is discussed in Appendix A.4.2.

**Remarks 2.2.** For diagnosing ND using resting-state fMRI, spatial models are more effective than sequential models. On ADNI and PPMI dataset, GSN and MLP outperform other spatial models, suggesting the superior isomorphism representation of GSN and the effectiveness of MLP for functional connectivity of this type of data. Among sequential models, Transformers yield good results across all three datasets, likely due to the self-attention mechanism, which can simultaneously

---

[4]Multi-class classification is shown in Appendix A.3.2

consider relationships between all elements in a sequence, particularly in longer sequences (such as those with up to 946 time points in ADNI)

## 5.4 Analysis on Neuropsychiatric Disorders Using Resting-State fMRI

Neuropsychiatric disorders are conditions that involve both neurological and psychiatric symptoms. Common neuropsychiatric disorders include Autism, Bipolar Disorder, Schizophrenia, Obsessive-Compulsive Disorder (OCD), and Attention-Deficit/Hyperactivity Disorder (ADHD). The following benchmark focuses on Autism classification using ABIDE data cohort. The comprehensive analysis of the ABIDE dataset is shown in Table 4. It indicates there is a performance difference between spatial and sequential models, where sequential models exhibit a slight performance advantage over spatial models in terms of better consistency in accuracy, precision and F1 score.

Table 4: Model performance on Neurodevelopmental Disease fMRI.

| ABIDE | | | | | | | | | | | | |
|---|---|---|---|---|---|---|---|---|---|---|---|---|
| | **Spatial Models** | | | | | | | **Sequential Models** | | | | |
| (%) | GCN | GIN | GSN | MGNN | GNN-AK | SPDNet | MLP | 1D-CNN | RNN | LSTM | Mixer | TF | Mamba |
| Acc | 62.93 ±6.65 | 55.61 ±1.85 | 54.93 ±4.89 | 55.71 ±3.01 | 53.56 ±2.63 | 68.20 ±2.87 | 58.83 ±0.90 | 67.41 ±2.03 | 54.15 ±0.98 | 57.66 ±1.67 | 57.37 ±3.24 | 62.93 ±2.20 | 66.62 ±1.09 |
| Pre | 58.18 ±16.75 | 63.49 ±6.03 | 35.07 ±14.67 | 60.94 ±8.11 | 33.76 ±12.08 | 68.08 ±2.65 | 59.01 ±1.01 | 67.93 ±1.77 | 56.05 ±4.96 | 58.65 ±1.04 | 57.81 ±2.98 | 63.21 ±2.23 | 67.53 ±1.46 |
| F1 | 59.37 ±13.19 | 45.64 ±7.24 | 41.88 ±11.21 | 50.31 ±6.21 | 40.24 ±8.03 | 68.03 ±2.72 | 57.72 ±1.88 | 67.11 ±2.12 | 44.84 ±1.93 | 55.12 ±3.72 | 54.91 ±5.34 | 61.78 ±3.10 | 66.47 ±2.20 |

However, it is important to underscore SPDNet, which consistently stays on the top of all evaluation metrics. We conducted a correlation analysis to examine the relationship between the accuracy of SPDNet and that of other models, with the results presented in Figure 4. The analysis shows that SPDNet's performance exhibits statistically significant differences from 9 of the other 12 models. The methodology innovations in SPDNet include two standout features: (1) maintaining the geometry of FC matrice through manifold-based feature representation learning, and (2) employing a spatial-temporal framework to capture dynamic patterns within evolving FC matrices. The top performance by SPDNet implies that the gateway to diagnose neuropsychiatric disorders might be a good spatial-temporal feature representation with great mathematical insight, which is supported by the biological evidence below.

**Remarks 3.1.** Autism and other neuropsychiatric disorders involve atypical neural connectivity (both increased and decreased variability in BOLD signals) and dynamics (alteration in the timing and coordination of neural activity) that affect social and cognitive processing (Müller & Fishman, 2011; Uher et al., 2014; Rudie & Dapretto, 2013; Menon, 2011; Just et al., 2012). Since the progression of autism impacts both network topology and neural activities, a spatial-temporal approach is more favorable for achieving promising diagnostic results.

**Remarks 3.2.** In Section 5.3, we conclude that spatial models outperform sequential models in early diagnosis of ND since the cognitive decline in ND is often associated with widespread neurodegeneration and impaired network functionality. Along with **Remarks 3.1**, converging evidence suggests that incorporating domain knowledge of the disease-specific pathophysiological mechanisms is crucial for method development and interpretation.

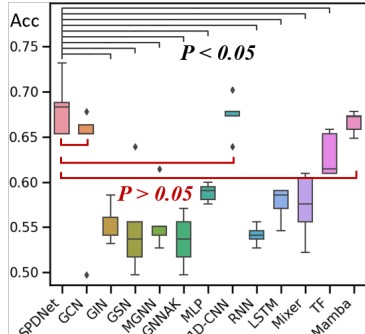

Figure 4: SPDNet exhibits significant performance gain over all other spatial models except for GCN.

## 6 Evaluate Model Explainability on fMRI Data

Here, we are interested in whether these models are interpretable in the neuroscientific sense. Specifically, we aim to determine whether the features considered critical by these models for prediction hold significance in the context of neuroscience. To achieve this, we conducted post-hoc analyses to

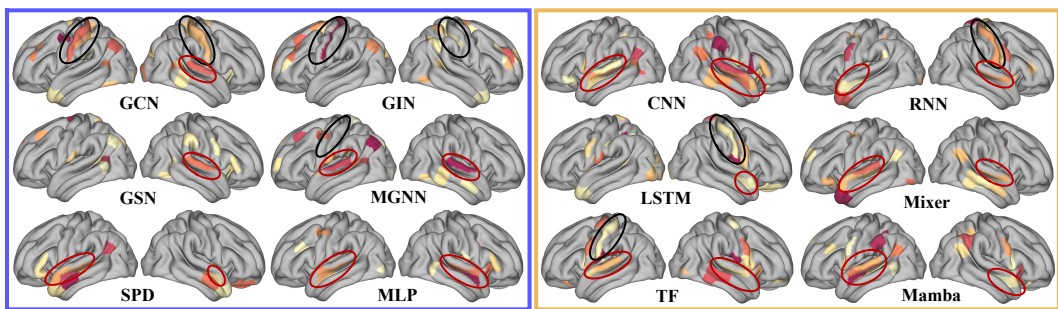

Figure 5: Brain mapping of global attention weights for features derived from spatial and sequential models. Black and red circles denote Motor and Language-related brain areas, respectively.

highlight the most salient features across different tasks and subsequently visualized these analysis on brain surface maps. This approach aims to validate whether the attention maps derived from different models are consistent with established neuroscience findings regarding task-specific brain activation patterns.

To generate an attention map that reflects the importance of specific regions-of-interest (ROIs), we introduce a learnable global attention assigned to each ROI. This attention is applied by multiplying it with the dimension of ROI of the initial input data, ensuring that the attention is directly associated with the input features. By applying attention prior to the data entering the models, we maintain a strict spatial correspondence between the attention map and the ROIs, preventing the possible misalignment caused by the non-linear operation in feature representation learning. The global attention vector is initialized with ones, and a *sigmoid* function is applied to constrain the attention weights between 0 and 1, representing the attention score for each ROI.

**Evaluation of brain attention maps on HCP task-specific data.** For brain mapping, we selected the top (critical) 30 brain regions with the highest corresponding weights. In Fig. 5, we present the brain mapping results from various deep models, including both spatial models (in blue) and sequential models (in orange). Black circles denote Motor-related regions, while red circles denote Language-related regions. *In spatial models*, GCN effectively identifies most motor-related brain regions, with the corresponding weights are relatively high (indicated by colors). GCN also identifies some language-related regions, though less comprehensively. Notably, MGNN selects more language-related regions by high probabilities, with the red circles indicating auditory association cortex areas. *Among sequential models*, the Transformer identifies more Motor-related brain regions, whereas the 1D-CNN identifies almost all the Language-related areas with relatively higher weights, indicating its emphasis on these regions.

**Remarks 4.1.** Figure 5 shows that while some models, such as GCN, can identify brain regions corresponding to specific tasks, they do not entirely align with the brain activation patterns established by neuroscience, and the selected brain regions appear relatively dispersed. We speculate that this is likely attributable to the following reasons:

1. Our analysis of the global attention weights revealed that their values were quite similar, leading to weak discrimination across ROIs. This may result in the failure to highlight the correct brain regions, even though it performs well in classification. This issue could be related to our initialization of global attention with all ones. When the input features exhibit inherent discriminability, the attention values tend to balance out, preserving the original discriminative relationships. To explore this further, we tried initializing attention weights randomly in the range [0, 1). While this approach promotes stability from initial randomness, the values remained fairly close, so the improvement in discrimination was limited.

2. Although the different models fit the data well and achieved high accuracy, the global attention maps generated by various models vary. This raises the question of whether the features highlighted by deep learning models align with those identified by neuroscience for specific tasks. Because the models may have learned task-specific patterns that are effective for classification but lack biological relevance. Reasons for this issue include (1) substantial

external noise in functional neuroimages, and (2) predominant inter-subject variations over subtle task-specific features.

To address the first challenge, we expect adding sparse constraint might be useful, however, it brings extra cost of computational complexity. Regarding the second issue, possible solutions include ensemble learning (to enhance the consensus) and dismantlement learning (to reduce the influence from noise).

**Remarks 4.2.** Although the prediction accuracy and the power of detect disease-specific brain regions are promising, the findings are not yet converging. One possible reason is the lack of standard attention module, integrated in the deep model, to estimate the importance of each brain region *on the fly*. A few deep models (Bedel et al., 2023) have emphasized the model explainalibity in the method design by reusing the existing attention components. However, it has not been fully investigated whether the learning mechanism of attention coefficients consistently maintains region-to-feature coherence (as explained in **Remarks 4.1**) throughout the entire deep neural network. To that end, further investigation into biologically-inclined feature representation learning and attention mechanisms is necessary.

## 7   DISCUSSIONS AND CONCLUSIONS

**Discussions.** We have collected sufficient supporting evidence to answer the questions in the beginning of Section 5. The quick answer to *(H1)* is "NO", based on the quantitative results shown in Tables 2, Table 3 and Table 4. Converging evidence shows that an in-depth understanding of the biological mechanism could significantly affect learning performance. Thus, the answer to *(H2)* is "YES" and the key is: integrating the neuroscience insight into model design which could not only lead to high accuracy but also uncover novel biological mechanisms using machine learning techniques. Another useful guideline for future fMRI-based deep models is: Explainable deep model, which is of high demand to jointly recognize cognitive tasks (or forecast disease risk) and yield task-specific (or disease-relevant) brain mappings. A successful explainable deep model will be very attractive in the neuroscience field given the surging number of neuroimaging data in many neuroimaging studies. Finally, the prediction accuracy for forecasting disease risks by current deep models is very promising. Considering the wide availability of MRI scanner in the clinic, there is substantial feasibility in implementing fMRI-based computer-assisted diagnosis systems for routine clinical practice. However, caution is needed due to the complex nature of disease progression and our incomplete comprehension of disease etiology.

**Conclusions.** We have conducted a comprehensive benchmark for current SOTA deep models in various fMRI studies. The strength of this work lies in a large scale of functional neuroimages with professional data pre-processing. We provide a set of useful guidelines of developing suitable deep models for new fMRI applications. Additionally, the pre-processed data is publicly available, fostering collaborative research between the fields of machine learning and computational neuroscience.

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

# A  APPENDIX

## A.1  DATA SOURCE AND DATA PREPROCESSING

### A.1.1  DETAILS OF DATASETS

**Human Connectome Project - Young Adult (HCP-YA).** The HCP dataset was collected using a Siemens 3T Skyra scanner with a customized 32-channel head coil from 1206 healthy young adult participants, ensuring high spatial and temporal resolution. Tasks included Working Memory, Gambling, Motor, Language, Social Cognition, Relational Processing, and Emotion Processing. Each task had two runs with alternating phase encoding directions. Key imaging parameters for the functional scans included a TR of 720 ms, TE of 33.1 ms, and 2.0 mm isotropic voxel size.

**Human Connectome Project in Aing (HCP-A).** The HCP-Aging dataset was collected using a Siemens 3T Prisma MRI scanner with a 32-channel head coil, capturing high-quality imaging data from older adult participants across a wide age range (36 to 100 years old). The dataset includes both structural and functional MRI scans, allowing for the study of age-related changes in brain structure and function. Three PsychoPy tasks covered in the HCP-A dataset include VisMotor Task, CARIT (Conditioned Approach Response Inhibition) Task and Facename Task. Imaging parameters for the functional scans include a TR of 800 ms, TE of 37 ms, and 2.0 mm isotropic voxel size, ensuring high spatial and temporal resolution for brain connectivity analyses.

**ADNI.** The ADNI dataset primarily uses 3T field strength scanners. It includes high-resolution 3D T1-weighted structural MRI and resting-state fMRI to capture resting brain activity. The protocols ensure standardized acquisition across sites, with quality control measures to maintain consistency at the Mayo Clinic. Pre-processing steps such as intensity normalization, gradient unwarping, and spatial normalization are applied to ensure reproducibility and reliability for longitudinal and cross-sectional Alzheimer's disease research.

**OASIS.** The OASIS dataset uses Siemens 3T MRI scanners with 16-channel head coils to achieve high-resolution imaging, typically with 1 mm isotropic voxel size for structural scans. The dataset includes T1-weighted, T2-weighted, and FLAIR structural MRI, along with BOLD functional MRI to assess brain activity during resting states. Comprehensive data pre-processing involves spatial and intensity normalization, alongside rigorous quality control to ensure high data quality. This dataset is important for longitudinal studies on aging and neurodegenerative diseases.

**PPMI.** The Parkinson's Progression Markers Initiative (PPMI) dataset employs advanced MRI scanners from GE, Philips, and Siemens, primarily using 3T field strength for high-resolution imaging. It includes high-resolution 3D T1-weighted structural MRI for anatomical mapping and resting-state fMRI to study functional connectivity. Data preprocessing involves intensity normalization, gradient unwarping, spatial normalization, and the use of Echo-Planar Imaging (EPI) for high temporal resolution. These protocols support robust longitudinal studies and cross-sectional comparisons in Parkinson's disease research.

**ABIDE.** The Autism Brain Imaging Data Exchange (ABIDE) has aggregated functional and structural brain imaging data to enhance our understanding of the neural bases of autism. All participants within ABIDE dataset were scanned on the same 3.0 Tesla Ingenia scanner with 15-channel head coil. ABIDE's comprehensive data supports detailed studies on brain functional connectivity and structural differences in ASD.

### A.1.2  PREPROCESSING

We independently processed the HCP[5], ADNI[6], and OASIS[7] datasets to obtain BOLD signals, employing a comprehensive preprocessing pipeline. The specific steps are described below. For the PPMI[8] and ABIDE[9] datasets, we used the preprocessed data provided by (Xu et al., 2024). For

---

[5] https://db.humanconnectome.org/
[6] https://adni.loni.usc.edu/data-samples/
[7] https://sites.wustl.edu/oasisbrains/
[8] https://www.ppmi-info.org/access-data-specimens/download-data
[9] https://fcon_1000.projects.nitrc.org/indi/abide/

detailed information on the preprocessing methodologies applied to these datasets, please refer to the original publication cited.

We used *fMRIPrep*[10] as the primary tool for preprocessing, given its capacity to provide a state-of-the-art, robust interface for functional magnetic resonance imaging (fMRI) data. *fMRIPrep* is designed to handle variations in scan acquisition protocols with minimal user intervention, ensuring high reproducibility and accuracy in preprocessing steps.

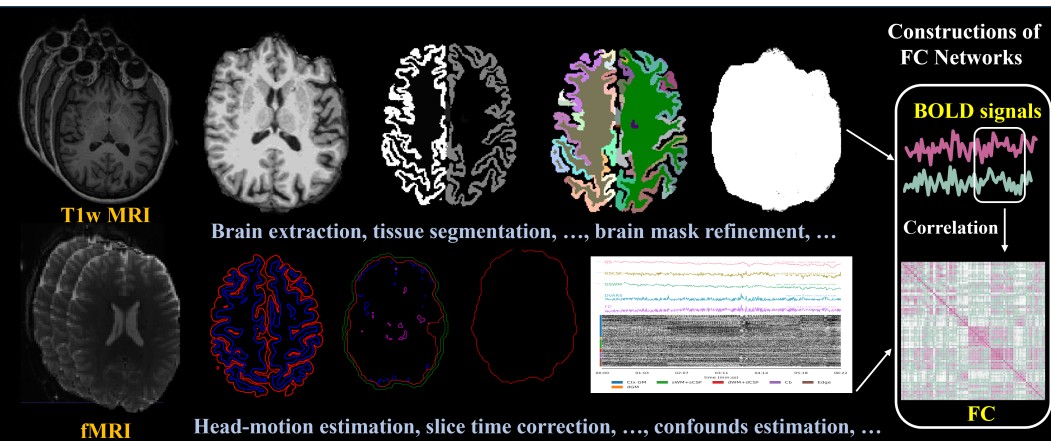

Figure 6: Workflow of constructing FC.

The fMRI data processing pipeline includes the following critical steps:

▷ *Structural MRI (T1-weighted) preprocessing*, which comprises:

- Brain extraction: Isolates the brain from the rest of the head in the MRI image.
- Tissue segmentation: Divides the brain into different tissue types (e.g., gray matter, white matter, cerebrospinal fluid).
- Spatial normalization: Aligns the brain images to a standard template for comparative analysis.
- Cost function masking: Masks non-brain areas to improve normalization.
- Longitudinal processing: Adjusts for within-subject variability over time in longitudinal studies.
- Brain mask refinement: Enhances the accuracy of the brain extraction process.

▷ *BOLD preprocessing*, which involves:

- Reference image estimation: Identifies a representative image from the BOLD series.
- Head-motion estimation: Detects and quantifies movements of the subject's head during scanning.
- Slice time correction: Adjusts for timing differences in slice acquisition within a single volume.
- Susceptibility distortion correction: Corrects distortions caused by magnetic field inhomogeneities.
- EPI to T1w registration: Aligns BOLD images to the anatomical T1-weighted image.
- Resampling into standard spaces: Converts images into standardized coordinate systems.
- Sampling to Freesurfer surfaces, HCP Grayordinates**: Projects the data onto standardized cortical surfaces for analysis.

---

[10]https://fmriprep.org/en/stable/

• Confounds estimation: Identifies and models sources of noise and artifacts.

▷ *Generation of BOLD time-series data for functional connectivity (FC) matrices construction*, facilitating subsequent analyses and ensuring comprehensive data preparation for downstream modeling tasks. This step involves extracting time-series data from the preprocessed BOLD images, which are then used to construct functional connectivity matrices, representing the temporal correlations between different brain regions.

## A.2 IMPLEMENTATION DETAILS

**Dataset-specific details.** For task-evoked fMRI, we typically take scan 1 as the training set, use 40% of scan2 for validation, and use the remaining 60% for testing. For the HCP-WM, additional processing steps were undertaken for subtask classification, which includes eight distinct subtask. Each complete scan was segmented into eight samples of 39 time points corresponding to these subtasks, with resting-state time points removed. We therefore expanded the size of the dataset to eight times that of the original Working Memory data.

For fMRI related to various diseases, given their limited data size, we employed cross-validation to train the models and calculated the average accuracy across all folds as the final performance metric. Specifically, for datasets with fewer than 500 samples, such as ADNI and PPMI, a 10-fold cross-validation was performed. Conversely, for datasets exceeding 500 samples, such as OASIS and ABIDE, a 5-fold cross-validation was performed. The parameter information of all models is shown in Table 5.

| (M) | HCP-Task | HCP-WM | ADNI | OASIS | PPMI | ABIDE |
|---|---|---|---|---|---|---|
| GCN | 1.79 | 1.79 | 0.38 | 1.38 | 1.29 | 1.29 |
| GIN | 3.89 | 3.89 | 1.11 | 4.37 | 4.33 | 4.32 |
| GSN | 0.92 | 0.92 | 0.698 | 0.738 | 0.695 | 0.696 |
| MGNN | 3.94 | 4.37 | 1.14 | 4.42 | 4.37 | 4.33 |
| GMM-AK | 290.3 | 290.3 | 290.3 | 290.3 | 290.3 | 290.3 |
| SPDNet | 0.19 | 0.19 | 0.049 | 0.064 | 0.067 | 0.059 |
| MLP | 66.9 | 33.46 | 3.536 | 13.32 | 7.084 | 7.083 |
| 1D-CNN | 2.22 | 2.22 | 0.716 | 1.602 | 0.754 | 0.753 |
| RNN | 1.19 | 3.89 | 3.38 | 3.48 | 3.27 | 3.27 |
| LSTM | 14.45 | 14.45 | 3.45 | 13.42 | 3.39 | 3.39 |
| Mixer | 6.78 | 1.65 | 0.782 | 1.63 | 2.19 | 3.297 |
| TF | 12.98 | 12.98 | 12.73 | 12.78 | 12.67 | 33.72 |
| Mamba | 27.05 | 27.05 | 6.84 | 26.84 | 26.79 | 26.79 |

Table 5: The number of parameters of the models on different datasets.

**Model-specific details.**

For all model training, we used the Adam optimizer with a learning rate of $10^{-4}$. The default hidden state dimension was set to 1024, except for ADNI, it only has 250 samples that belong to 2 classes, the large number of hidden dimension will lead to overfitting, so we set it to be 512.

We do zero-padding to make all the BOLD sequences within the same dataset have the same length when perform sequential models. Specifically, we identify the maximum number of time points $T_{max}$ present in the longest BOLD signal within the dataset. For BOLD signals with fewer than $T_{max}$ time points, we append a zero matrix $\mathbf{O} \in \mathbb{R}^{N \times (T_{max} - T_d)}$, where $T_d$ is the number of time points for $d^{th}$ data cohort. This padding ensures that all BOLD matrices within the dataset are of consistent size, facilitating their use in sequential models.

*GCN and GIN*. We used two pure graph convolution layers in the models without any residual connections.

*GSN*. We used the default parameters specified in the original paper, with the exception of setting 'd_out' to 256.

*Moment-GNN.* We calculated topological features (K=10, dimension 45) from the adjacency matrix based on the original code and concatenated them with the correlation vector as final node features, using the GIN model as backbone as recommended in the original paper.

*GNN-AK.* We used all default settings for the ZINC dataset. We selected GINEConv with two mini layers in this model.

*SPDNet* was used with all default settings from the original paper, and we adjusted model with diffident input dimensions according to different datasets.

*MLP.* we took the lower triangle of the original functional connectivity matrix and flatten it into a vector, which was then input into a three-layer MLP for training.

*1D-CNN* was constructed based on the method described in (El Gazzar et al., 2019), comprised two layers of 1D convolution with BatchNorm.

*RNN and LSTM.* We used two layers of RNN/LSTM and employed the final hidden node for prediction, the classification head has two Linear layers.

*MLP-Mixer* had a token embedding dimension of 256 or 512, depending on the temporal length of the sample, and a channel embedding dimension of 1024, with a total of four layers.

*Transformer.* We used four layers of Transformer encoder layers with either two or four heads, incorporating position encoding as outlined in the original paper. And the hidden state is 512 for task fMRI for better classification.

*Mamba.* We averaged the last hidden layers of the original Mamba model and fed them into a new classification head with a 'd_model' of 1024, maintaining four layers in total while keeping other settings at their defaults.

In addition, our experiments are run on a local platform that has dual Intel(R) Xeon(R) Gold 6448Y CPUs and four NVIDIA RTX 6000 Ada GPUs.

### A.3 MORE EXPERIMENTS

#### A.3.1 HISTGRAM DISTRIBUTIONS OF MODELS OVER ALL THE DATASETS

We plot the KDE curve, which offers a smooth, continuous estimate of the data's probability density function by applying a kernel function (often Gaussian) to each data point and summing the results, for sequential model (in yellow) and spatial model (in blue) in Fig. 7 where horizontal and vertical axes represents the equal intervals (bins) of accuracy and the ratio of the number of data points in each bin respectively. This figure shows the performance distribution of spatial and sequential models across various datasets. The distinct distribution patterns vividly illustrates their differences.

Table 6: Multi-class Classification on ADNI.

| ADNI | | | | | | | | | | | | |
|---|---|---|---|---|---|---|---|---|---|---|---|---|
| | Spatial Models | | | | | | Sequential Models | | | | | |
| (%) | GCN | GIN | GSN | MGNN | GNN-AK | SPDNet | MLP | 1D-CNN | RNN | LSTM | Mixer | TF | Mamba |
| Acc | 50.00 ±6.51 | 51.60 ±5.20 | 52.80 ±5.31 | 48.80 ±5.31 | 52.40 ±6.56 | 52.40 ±5.20 | 46.40 ±7.42 | 46.00 ±5.44 | 45.60 ±6.25 | 46.00 ±7.43 | 48.40 ±4.18 | 52.00 ±6.93 | 47.20 ±6.14 |
| Pre | 36.08 ±14.22 | 39.75 ±13.50 | 53.23 ±10.33 | 40.58 ±10.28 | 46.07 ±9.48 | 37.01 ±8.89 | 46.52 ±12.03 | 36.40 ±9.72 | 40.95 ±11.29 | 25.89 ±11.28 | 48.06 ±12.84 | 47.63 ±19.50 | 38.55 ±13.23 |
| F1 | 38.49 ±9.73 | 41.76 ±7.65 | 48.21 ±7.48 | 38.71 ±6.11 | 43.20 ±6.55 | 31.63 ±8.76 | 43.83 ±9.13 | 39.21 ±7.71 | 39.24 ±7.14 | 31.87 ±9.93 | 39.40 ±5.47 | 44.03 ±11.32 | 37.19 ±5.53 |

#### A.3.2 ADDITIONAL EXPERIMENTS ON ADNI

We also implemented 4-class classification on ADNI. ADNI contains 5 detailed classes: CN (clinically normal), SMC (subjective memory concerns), EMCI (early mild cognitive impairment); LMCI (late mild cognitive impairment) and AD (mild Alzheimer's disease dementia). For the 4-class classification, we merged CN and SMC into one class because they have similar patterns. The results are shown in the Table 6. Among the spatial model, GSN demonstrates superior performance with an accuracy of 52.8%. GNN-AK and SPDNet follow closely, each with an accuracy only 0.4% lower

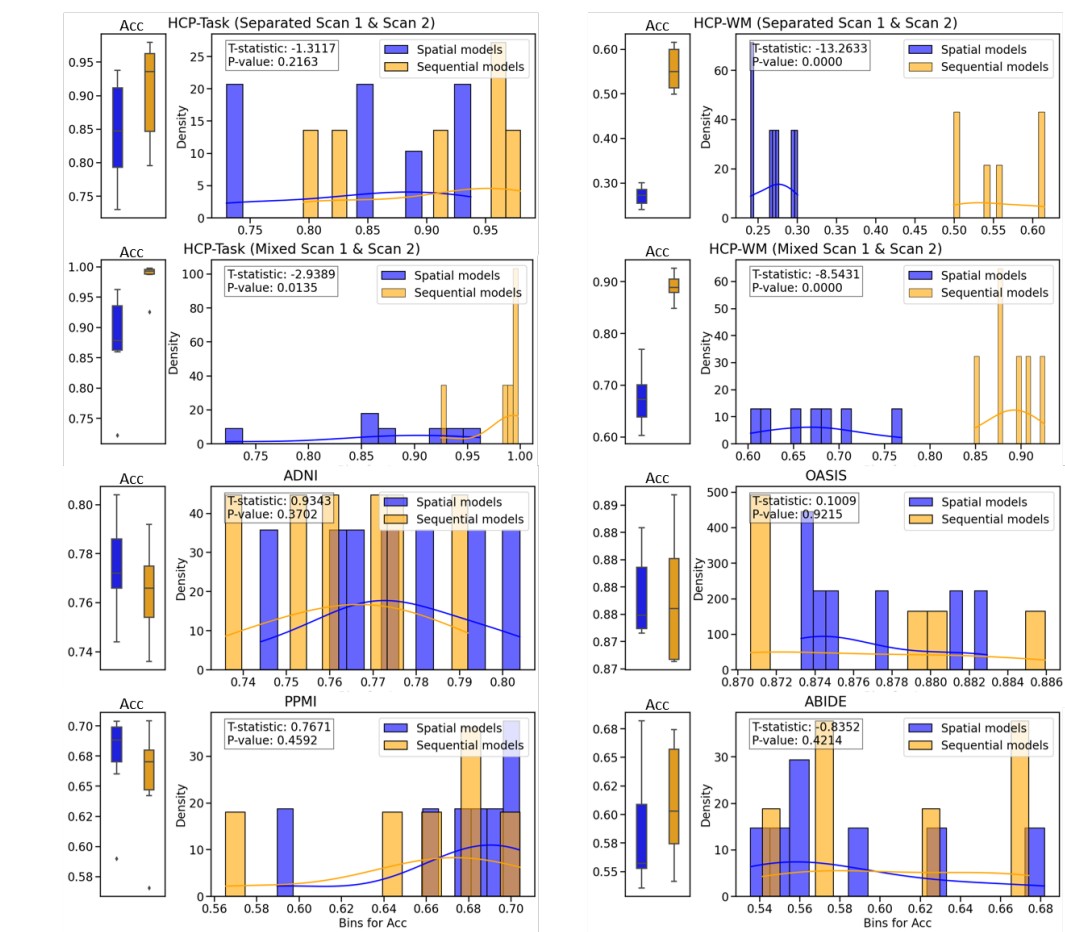

Figure 7: The histograms and distribution fittings for the accuracy that derived from different datasets.

than GSN. Among the sequential models, the Transformer outperforms other models, achieving an accuracy of 52%. Notably, spatial models generally exhibit superior performance compared to sequence models, which aligns with the trends observed in the results from the 2-class classification scenario.

### A.4 EVIDENCES SUPPORT REMARKS

#### A.4.1 REMARKS 1.2.

The analysis suggests that sequential models are more effective in task-evoked fMRI classification. Meanwhile, there exist significant variations of learning performance within spatial and sequential models.

Specifically, SPDNet achieves the highest accuracy in spatial models, with 93.76% in the Separated Scan setting and 96.23% in the Mixed Scan setting on HCP-Task and 76.91% in the Mixed Scan setting on HCP-WM. This can be attributed to SPDNet's capability to effectively handle high-dimensional manifold data, which is beneficial for capturing the spatial relationships within high-resolution fMRI data. GSN is the second top-ranked spatial model, with accuracies of 88.78% in the Separated Scan setting and 92.54% in the Mixed Scan setting on HCP-Task, and achieving accuracies of 30.06% in the Separated Scan setting and 71.18% in the Mixed Scan setting on HCP-WM. The innovation of GSN lies in its ability to model sub-structural patterns, such as fixed-length cycles or cliques. This approach enhances expressivity by effectively incorporating the counting of sub-graph isomorphisms. GCN and GIN also show reasonable performance, with GIN slightly outperforming GCN due to its enhanced graph isomorphism consideration. MomentGNN and GNN-AK demonstrate lower

performance, suggesting that MomentGNN's designation of capturing statistical moments of graph features and GNN-AK's use of a GNN as a kernel to encode local sub-graphs might not be as effective for task-specific BOLD signal variations as their showcase in molecular graphs.

On the other hand, sequential models such as 1D-CNN, LSTM, Transformer, and Mamba demonstrate outstanding performance. Notably, 1D-CNN achieves the highest accuracy of 97.96% in the Separated Scan setting on HCP-Task due to its effectiveness in capturing temporal patterns through convolutional operations. Additionally, MLP-Mixer attains the highest accuracy of 99.81% in the Mixed Scan setting of HCP-Task because of its ability to represent information in both channel (ROI) and temporal dimensions. LSTM also performs exceptionally well, achieving 99.51% in the Mixed Scan setting on HCP-Task and 61.46% in the Separated Scan setting on HCP-WM, thanks to its design for handling long-term dependencies in temporal dynamics. The Transformer, with its attention mechanisms, excels in capturing global dependencies across time in task fMRI data.

### A.4.2    REMARKS 2.1.

Convergent evidence shows that (1) deterioration of brain function commences several years prior to the cognitive decline and (2) the prodromal period can last years or even decades before the clinical diagnosis is made (Viola et al., 2015; King et al., 2009). The benchmark results in Table 2 bottom and Fig. 3 provide a solid evidence that current deep models have the high potential to deploy to the clinical routine of early diagnosis of ND. For example, GSN achieves the highest accuracy (70.40% ± 12.48) among spatial models. GIN also performs well, with consistent accuracy and precision. Among sequential models, the Transformer stands out with the highest accuracy (70.43% ± 11.74) and robust F1-score.

