# OpenReview forum: "Machine Learning Benchmark on Dynamic Functional Connectivity: Promise, Pitfalls, and Interpretations"
_ICLR.cc/2025/Conference — Submitted to ICLR 2025_

### Official Review · Reviewer_i3zP · 2024-10-28

**Soundness:** 2
**Presentation:** 3
**Contribution:** 2
**Rating:** 3
**Confidence:** 3

**Summary:**

This paper proposes a novel benchmark that utilizes an unprecedented amount of existing task-evoked fMRIs and disease-related rs-fMRIs together to provide a well-founded empirical guideline through comprehensive and consistent performance comparisons by dividing them into spatial and sequential models. The authors analyze the performance of SoTA methods on cognitive task recognition and disease diagnosis using fMRI and provide guidelines for model design in each scenario. They also conduct a comprehensive evaluation and statistical analysis to examine model performance consistency across datasets, identify brain regions related to model features via attention-based brain mapping, and highlight the need for a standardized attention module.

**Strengths:**

1. Unlike the existing benchmarks, the authors composed task-fMRI datasets consisting of various cognitive tasks and rs-fMRI datasets related to disease diagnosis, and provided a comprehensive summary the results after evaluating various deep learning methods on the composed datasets.
2. The authors conducted performance comparisons across groups with statistical analyses to identify which method groups performed significantly better on specific datasets using significance tests.
3. The authors provided visualization of brain activation through global attention weight to assess model explainability.

**Weaknesses:**

1. While it is valuable to combine task-fMRI and rs-fMRI datasets as a benchmark to gain insights into deep learning model design in neuroimaging, the benchmark dataset lacks some novelty, as it primarily consists of datasets previously introduced in studies such as BrainGB (Ref1) and NeuroGraph (Ref2).

- Therefore, it would be more differentiated if the models of each group were evaluated in each scenario and with each setting through a table.
- It would be helpful to include a detailed explanation of the criteria for the configuration of the input data of the deep learning model in using the benchmark, such as performance comparison of various input feature types in FC network data or network topology configuration.
- An additional explanation on how dynamic functional connectivity is configured would be beneficial. For example, this could involve the window size (which determines the number of dynamic FC matrices).
2. The models are referred to as SoTA deep learning models, but they may not demonstrate optimal performance, as it lacks graph models (such as BrainGNN-Ref3) and transformer models (Ref4) that have shown strong performance on specific datasets. Additionally, computational details such as model parameters, memory usage, and training time are challenging to compare across models.
- It would be helpful to provide an explanation of the selection criteria, as the model structures used appear relatively simple
- Including an additional table to show the memory usage, training time, and parameters of spatial-based and sequential-based methods would facilitate an intuitive understanding of the differences between models.
3. Although model global attention weights were used to assess explainability through brain activity maps, attention maps specific to neurodegenerative or neuropsychiatric disorders were not provided separately, and a standardized attention module was lacking. BrainGNN [ref3] evaluated model explainability in disease diagnosis based on rs-fMRI data. If a similar approach was applied in this study, additional analysis on why it may not have performed as expected would be require.
4. The authors performed statistical analysis between groups with significant test in figure3, it is difficult to identify intuitive individual performance comparison of the models in the experimental table. it seems that further performance comparison of the models is needed for other datasets like figure4.

**Questions:**

1.	Could you clarify if the dynamic functional connectivity matrix is constructed by dividing the RoI signal with a fixed window size? If so, it would be helpful to see a baseline or performance comparison for different window sizes.
2.	Additionally, I understand that the dynamic FC matrix generates multiple FC matrices through the window mechanism, unlike the static fc map that performs pearson correlation over the entire RoI signal, if this is correct, could you explain how these dynamic FC matrices are formatted for the model input?
3.	Could you explain on the criteria for selecting spatial and sequential models? Specifically, I am curious if you chose the models based on highest performance metrics or if commonly used architectures were tested.
- For instance, did you conduct comparison experiments with baseline models like brainNetCNN (Ref5), which uses a specific kernel shape, also BrainGNN (Ref3) and Transformer (Ref4)?
4.	In Remarks 1.1, I understood that the reason why sequential models performed well on the task fMRI dataset is that methods that directly use the BOLD signal as input data can capture dynamic variability better than dynamic FC networks that are generated based on a specific window. Is this correct? If so, could you explain on the variability differences between task-fMRI and rs-fMRI?
5.	Could you explain in more detail how the global attention weight is updated in Section 6? In the paper, I understand that the attention maps are mainly capturing on motor-related regions or language-related regions, but there are 7 tasks in HCP. do you have brain mapping results for other regions?
6.	I was unable to find a link to the code in the paper. If possible, could you provide to the code?


Reference:

Ref1. BrainGB: A Benchmark for Brain Network Analysis with Graph Neural Networks (TMI 2022)

Ref2. NeuroGraph: Benchmarks for Graph Machine Learning in Brain Connectomics (NIPS 2022)

Ref3. BrainGNN: Interpretable Brain Graph Neural Network for fMRI Analysis (Medical image analysis, 2021)

Ref4. BRAIN NETWORK TRANSFORMER (NIPS 2022)

Ref5. BrainNetCNN: Convolutional neural networks for brain networks; towards predicting neurodevelopment (NeuroImage, 2017)

---

### Official Review · Reviewer_vroc · 2024-11-01

**Soundness:** 2
**Presentation:** 2
**Contribution:** 3
**Rating:** 5
**Confidence:** 4

**Summary:**

This paper aims to establish a well-founded empirical guideline for designing deep models for functional neuroimaging by integrating methodological principles with knowledge from the neuroscience domain. Specifically, the authors constructed, pre-processed, and released a diverse set of datasets across various tasks for the fMRI-based benchmark.

**Strengths:**

1.	By processing a diverse range of datasets and establishing benchmarks for multiple tasks, the authors enable researchers to effectively assess and compare the performance of various algorithms, models, or systems. This standardized testing approach facilitates the identification of the best-performing methods for specific tasks.
2.	The authors evaluated the performance of several state-of-the-art models on different tasks and provided a comprehensive analysis of the experimental results, demonstrating that selecting the appropriate method is essential for achieving optimal results across varying tasks.
3.	The use of neuroimaging knowledge to interpret performance differences among models introduces a novel perspective, which could offer valuable inspiration for future research directions.
4.	The authors proposed a post-hoc method to create task-specific attention maps and evaluated neural activation patterns across various tasks. This approach seeks to reveal neurobiological mechanisms using data-driven methods, which is crucial for advancing machine learning in computational neuroscience and clinical applications.

**Weaknesses:**

1. The paper presents some confusion regarding the concept of dynamic functional connectivity.
2. This paper summarizes existing datasets. However, further depth is needed regarding the related methods and quality control of the datasets.
3. The authors summarized seven datasets. However, some other important and commonly used datasets, such as ABIDE II, Cleveland Neuropsychiatry Study (CNP), The Pediatric Obsessive-Compulsive Disorder Treatment Study (POND), and NKI-Rockland Sample, seem to be omitted.
4. From a methodological perspective, the authors only considered deep learning methods, overlooking traditional machine learning approaches and other methods, such as deep forest.

**Questions:**

1. The paper raises questions about dynamic functional connectivity. In the abstract and introduction, the authors refer to the dynamicity of functional connectivity, rather than the dynamic functional connectivity as defined in neuroscience. In the Abstract and Introduction, the definitions of the core concepts such as brain networks, structural functions, functional effective, dynamics, and non-dynamics are not mentioned, making the overall research subject appear unclear.

2. The author should attempt to utilize more dynamic functional connectivity methods, particularly in the two components of state number estimation and connectivity learning.

3. In this paper, the differences and core issues between dynamic functional connectivity and functional connectivity seem to be inadequately addressed. Especially in the experiments, there is a focus on comparing classification performance, lacking an analysis of dynamic network connectivity.

4. The authors should consider incorporating more important datasets and other types of brain functional connectivity methods.

---

### Official Review · Reviewer_KpTp · 2024-11-01

**Soundness:** 3
**Presentation:** 2
**Contribution:** 1
**Rating:** 3
**Confidence:** 3

**Summary:**

The authors apply a variety of classification models to BOLD signal time-series, and functional connectome data from a collection of large public datasets to establish a benchmark of how well current models are performing, and which types of models perform best. The classification tasks include identifying the task being performed by the patient under fMRI imaging, and classifying a patient as being in a disease or healthy group. The datasets include the Human Connectome Project (HCP), Alzheimer's Disease Neuroimaging Initiative (ADNI), Open Access Series of Imaging Studies (OASIS), Parkinson’s Progression Markers Initiative (PPMI), and Autism Brain Imaging Data Exchange (ABIDE), some of which needed to be preprocessed to generate time series or connectome data. The models either operate on time series data ("sequential models") or connectome data ("spatial models") and include CNNs, RNNs, GNNs, and transformers. One of their central conclusions is that sequential models perform better for task classification because tasks are dynamic activities while spatial models perform better for disease classification since disease is a more slowly evolving state.

**Strengths:**

The authors present a well-designed set of experiments comparing classification performance of various models that operate on either time-series, or connectome data. They use well-known models and well-known datasets so their results are meaningful to the fMRI field.

**Weaknesses:**

The first claimed contributions is a collection of preprocessed datasets, however this work uses existing image data and existing preprocessing algorithms. Further, some of the datasets have already been preprocessed, and there is little to no description of how the data will be released. All of limits the value of their data contribution.

The authors present a dichotomy between "spatial" and "sequential" data. However, unless I am misunderstanding their setup, there is a key relationship between these two forms in that the spatial data can be written as a function of the sequential data i.e. the sequential data has at least as much information as the spatial data. In theory, a good sequential classifier could always outperform a spatial model. I believe that the main results in this paper are not biological but a form of the classic bias-variance tradeoff.

The authors aim to validate models by comparing a set of attention maps to the "brain activation patterns established by neuroscience," but there is little description of what activation patterns they are referencing. Figure 5 shows a set of outlines that vaguely identify "Motor and Language-related brain areas" but these outlines vary from panel to panel, and it seems that their black outline follows the postcentral gyrus, which is generally associated with somatosensory, not motor activity.

One of their central conclusions is that spatial models are better for disease classification and sequential models are better for task classification. There are already ample results on these setups (e.g. Wang et al. 2020 for task classification with fMRI time-series or Dadi et al., 2019 for disease prediction with functional connectomes), so I am not sure their conclusion is surprising. They also state that disease "forecasting" is promising, but it seems all their experiments involve classifying patients with an existing disease, not forecasting future disease.

Misc.
- There are several typos or grammatical errors: "Since connectome is often encoded...", "As shown in Table 1, we uses a diverse...", "Human Connectome Project in Aing"
- The equation W=XX^T only works if X is normalized appropriately.
- The explainability section mentions both interpretability and explainability while there is a difference between these concepts, see Rudin, 2019.

**Questions:**

What is the meaning of the +/- range in Table 3?

**Details Of Ethics Concerns:**

I originally swapped this review with another review. There are no ethics concerns on this submission.

---

> ### Comment · Reviewer_KpTp · 2024-11-12
> **Swapped reviews**
>
> Hi - I am so sorry but it appears I have swapped my reviews of two papers, so please disregard this for the moment. Trying to rectify this ASAP.

---

> > ### Comment · Reviewer_KpTp · 2024-11-12
> > **Swapped reviews (resolved)**
> >
> > The review should be correct now.

---

### Official Review · Reviewer_N285 · 2024-11-04

**Soundness:** 1
**Presentation:** 1
**Contribution:** 2
**Rating:** 3
**Confidence:** 5

**Summary:**

The authors implemented a variety of benchmarking machine learning models for functional Magnetic Resonance Imaging (fMRI) data analysis with conducting comprehensive evaluation and statstical analysis on seven publicly available fMRI datasets. Based on the results, the authors tried to provide some guidance on developing appropriate models for new fMRI studies, such as which model achieve SOTA performance and which model types are favorable for task-evoked/resting-state fMRI analysis, respectively.

**Strengths:**

This study addresses the important issue of adopting an appropriate deep learning model tailored to the characteristics of each fMRI study.
The authors implemented several different models on both task-evoked and resting-state fMRI data and compared the results, providing insights into performance from different perspectives.

**Weaknesses:**

In implementing multiple models, key hyperparameters that can significantly impact model performance are not properly optimized.
For example, in graph-based models, performance can vary greatly depending on how the graph topology is consturcted for each study, but the authors adopted the simplest method of remaining top 10% of values in FC matrix. In addition, other model hyperparameters, such as the depth of the layers and the number of nodes within the layer, were also applied to the default settings provided in each original paper with fixed learning rate of 0.0001. Since these hyperparameters can greatly affect performance depending on the size of the data and the difficulty of the task, it is common to optimize them based on validation data for a fair comparison. The reviewer believes that this limitation is closely related to the fact that even the simplest models, such as MLP, result in high performance with little influence on hyperparameter settings.

The authors stated in their paper that they compared SOTA methods, but this claim is difficult to accept since most implementation models are traditional and basic. In particular, the authors implemented spatial and sequential models separately for fMRI analysis, but many recent studies are simultaneously considering spatial-temporal features of fMRI.

**Questions:**

Given that hyperparameter optimization is essential for a fair comparison of model performance, why did the author not take this into proper consideration? In particular, constructing the graph topology plays a key role in graph-based models, but the authors constructed it based on a fixed threshold (e.g. top 10%) across all datasets. Furthermore, for a fair performance comparison, model hyperparameters—such as learning rate and layer depth, which significantly impact each model's convergence during training—should be adaptively set based on the model's size and task difficulty. However, all models in this study were assigned a fixed learning rate of 0.0001 and used the default architectural settings from their original papers. This reviewer believes that, at a minimum, model hyperparameters should be optimized based on the validation set in cross-validation for a fair comparison instead of adopting the default settings for each model.

Although this paper claims to implement current SOTA models, most of them are traditional and fundamental approaches. Furthermore, while many models that integrate spatial and temporal features have been proposed for fMRI data analysis, this paper only implements models that treat these features separately.

---

### Meta-Review · Area_Chair_mNDa · 2024-12-11

**Metareview:**

This submission contributes a benchmark of dynamic functional connectivity methods on a large dataset. It however failed to convinced the reviewers that it meets the high bar of ICLR. A variety of limitation were put forward, most strikingly the incomplete baselines and no hyper-parameter optimization, in particular for constructing the graph topology.

**Additional Comments On Reviewer Discussion:**

There was no discussion as the authors did not submit a rebuttal.

---

### Decision · Program_Chairs · 2025-01-22

Reject